# Influence of Nanoscale Textured Surfaces and Subsurface Defects on Friction Behaviors by Molecular Dynamics Simulation

**DOI:** 10.3390/nano9111617

**Published:** 2019-11-14

**Authors:** Ruiting Tong, Zefen Quan, Yangdong Zhao, Bin Han, Geng Liu

**Affiliations:** 1Shaanxi Engineering Laboratory for Transmissions and Controls, Northwestern Polytechnical University, Xi’an 710072, China; binhan@mail.nwpu.edu.cn (B.H.); npuliug@nwpu.edu.cn (G.L.); 2Shanghai Aircraft Design and Research Institute, Shanghai 200436, China; quanzefen@163.com; 3Aerospace System Engineering Shanghai, Shanghai 201109, China; npuzhaoyangdong@163.com

**Keywords:** textured surface, subsurface defect, friction, molecular dynamics simulation

## Abstract

In nanomaterials, the surface or the subsurface structures influence the friction behaviors greatly. In this work, nanoscale friction behaviors between a rigid cylinder tip and a single crystal copper substrate are studied by molecular dynamics simulation. Nanoscale textured surfaces are modeled on the surface of the substrate to represent the surface structures, and the spacings between textures are seen as defects on the surface. Nano-defects are prepared at the subsurface of the substrate. The effects of depth, orientation, width and shape of textured surfaces on the average friction forces are investigated, and the influence of subsurface defects in the substrate is also studied. Compared with the smooth surface, textured surfaces can improve friction behaviors effectively. The textured surfaces with a greater depth or smaller width lead to lower friction forces. The surface with 45° texture orientation produces the lowest average friction force among all the orientations. The influence of the shape is slight, and the v-shape shows a lower average friction force. Besides, the subsurface defects in the substrate make the sliding process unstable and the influence of subsurface defects on friction forces is sensitive to their positions.

## 1. Introduction

With the development of micro/nano electromechanical systems [1], the sizes of components come to a micro/nanoscale, and the molecular forces between the contact bodies play an important role. It is inevitable that there are some defects in the materials, and the defects in the surface or the subsurface will influence friction behaviors. In the work of Qi et al. [2], just 1Å (10^−10^ m) of surface roughness could affect the coefficients of friction (COFs) of clean, incommensurate Ni interfaces. The surface or the subsurface defects could influence the surface roughness or the molecular forces, while the detailed friction behaviors between nanoscale contact bodies, considering the surface or subsurface defects, are still unknown.

Generally, the profile of a contact surface is a random one, and the random profile is always simplified as ideal cylinder or sphere asperities. In a nanoscale sliding contact, the asperities on the surface can help to reduce contact area and surface-to-volume ratio, which will reduce friction force further. In addition, the artificial nanoscale textured surfaces have proved that they could greatly reduce adhesion and friction of many materials [3,4,5,6]. At nanoscale, the spacings between textures could be seen as artificial defects, which play the main role in reducing contact area. The possible parameters which influenced the friction properties were investigated [3,4,5,6].

Many experiments were performed to investigate the friction properties of nano-textured surfaces. An early study [7] showed that the nanoscale grain holes on the polysilicon surface could help to reduce the real contact area and sticking, and enlarging the hole size induced a further reduction of sticking. Zou et al. [8] compared the adhesion and friction performances of a silica nanoparticle-textured (SNPT) surface with a silicon oxide films surface, and the improved adhesion and friction performances of the SNPT surface were attributed to its reduced contact area. Nair and Zou [9] investigated the friction behaviors of nano-textured samples fabricated by aluminum-induced crystallization of amorphous silicon. Compared with the smooth surface, nano-textured surfaces reduced the friction forces greatly, and a higher texture height induced lower friction forces. Zhao et al. [10] prepared Au surfaces with micro/nano hierarchical structures and investigated the effects of textures on nanotribological properties. The nanotribological performances could be greatly improved by designing a suitable surface topography, and the friction force decreased as cylindrical pillar area density and height increased. Zhang et al. [11] found that Au nanoparticle-textured surfaces with a higher texture height could reduce the friction forces more efficiently. Yu et al. [12] studied the friction behaviors of the surfaces with grooves at the nano-, micro-, and macroscales. The materials included copper alloys, 52,100 steel, tungsten carbide, and silicon stamps, and the sliding direction played a key role in affecting the COFs. Wu et al. [13] used the surface of the fluorine-doped tin oxide wafer as a nanodot-textured surface, and they also prepared the nanorod-textured surface and nanocomposite-textured surface to investigate the effects of different shapes. The nanorod-textured surface showed better adhesion and friction performances. In the work of Steck et al. [14], a novel strategy for fabricating deformation-resistant nano-textured surfaces was developed, and the IP-DIP/Al_2_O_3_ core-shell nanostructures showed 85% lower COFs than the bare IP-DIP nanodots.

The experiments have obtained lots of valuable results, and numerical simulations on the nanoscale friction behaviors were also performed, of which molecular dynamics (MD) simulation was the most widely used one. Zhang et al. [15] found that the surface roughness tended to increase kinetic friction between Al–Al or α-Al_2_O_3_/α-Al_2_O_3_ interfaces. Chen et al. [16] performed MD simulations to investigate a Fe tip sliding on a Fe substrate with textured surface. The textured surfaces showed lower friction forces than a smooth surface, and a surface with parallel grooves presented lower friction than that of a rectangular-dimpled textured surface. Li et al. [17] studied nanoscale high-speed grinding behaviors of Cu by examining the effects of texture density, texture direction and texture shape. The surface with a lower texture density reduced the grinding forces and average COFs effectively, while the influence of texture orientation was slight, due to a high cutting depth. Santhapuram et al. [18] performed MD simulations on the friction properties of an aluminum surface with cylindrical and spherical textures, and the COFs depended on texture shape and tip radius. Using a multiscale method, Tong et al. [19,20] investigated the friction behaviors of the sliding contacts between rigid tips and textured surfaces, and a higher texture height and larger texture spacing could reduce the friction forces for the rectangular textures, but the model was a two dimensional one. In further work, they performed coarse-grained MD simulations on the friction behaviors of the textured Ag coatings [21]. The friction properties were influenced by the texture width, depth, orientation and shape, and the textured surface with 0° orientation and a rectangular shape showed lower average friction forces. For a textured gold film, the average friction forces and the surface temperature of the substrate were decreased when increasing the texture depth [22].

There are always defects in materials, due to various reasons, which change the properties of the materials. Some defects are beneficial to materials, and point defects increase the conductivity of a semiconductor material. On the other hand, the defects may play a negative role; the dislocation defects made the material easy to fracture and reduced the thermal conductivity of single-walled carbon nanotubes [23,24]. Due to the existence of defects, there were more cracks and local stress concentration points, which greatly reduced the mechanical properties of graphene. Meanwhile, the fracture strength was reduced, due to local stress points caused by defects [25]. For a carbon honeycomb with a single atom vacancy, the tensile strength was decreased by 11.5% in the cell axis direction, but only 5% in the other two directions. The position of vacancy also affected the mechanical properties in cell axis direction [26]. Kopta et al. [27] performed AFM experiments between Si-tips and the mica, and the results showed that the rupture of the Si–O bond might produce defects. When defects were accumulated beyond a critical concentration, they were grown to wear, which had a noticeable contribution towards friction. From the experiments of Ref. [28], the authors found that the vacancies increased the friction in graphene significantly. Guo et al. [29] carried out molecular-force-field calculations to study the interlayer friction between graphene sheets with defects. They found that the change of interlayer distance and the defects might provide a way to realize ultralow friction and control the friction properties in graphene sheets. Using MD simulations, Liu et al. [30] studied the friction characteristics of a nanotube and a graphene. They placed a vacancy defect at the center of a single-layer graphene, and a high positive lateral force was presented when the tip approached the defect. Sun et al. [31] investigated the friction behaviors of a diamond tip sliding over a graphite by MD simulations. For defective graphite with a defect in the surface, a single vacancy in the interior layer decreased the COFs. For the metal materials, Chen et al. [32] developed an MD model to study the effects of vacancy defects on the friction forces of the single crystal iron, and the average friction forces increased with the defect concentration.

From the references above, the textured surfaces or the surface defects influence the friction behaviors greatly, while the effects of the subsurface defects on friction properties are still unknown. Besides, the detailed influence of the texture parameters on the friction process needs further study. Therefore, in this paper, textured surfaces with different depths, widths, orientations, and shapes are modeled, and the friction behaviors between a rigid cylindrical tip and the textured surfaces are investigated. Furthermore, the influence of subsurface defects in the substrate is also discussed.

## 2. Model Description

In this paper, an MD model is used to study the influence of nanoscale textured surfaces on the friction behaviors of the sliding contacts. As shown in Figure 1, the MD model consists of a rigid cylindrical tip and an elastic substrate. The simulation parameters are shown in Table 1. The material of the tip and substrate is single crystal copper. The tip consists of 5504 atoms and its radius is *R* = 2.169 nm. The dimension of the substrate is 21.69 nm × 5.784 nm × 7.23 nm, in *x*, *y* and *z* directions, respectively, and there are 78,720 atoms. The number of atoms in the substrate will be varied slightly because of different textures. All simulations are performed using Large-Scale Atomic/Molecular Massively Parallel Simulator (LAMMPS) [33] and OVITO is used to visualize the results [34].

There is no thermal motion of the atoms in the cylindrical tip, and the tip can be regarded as a rigid one. Periodic boundary conditions are applied in *x* and *y* directions to reduce the size effects and CPU cost. Non-periodic boundary is used in *z* direction. The substrate is composed of three parts: boundary atoms, thermostat atoms and Newtonian atoms. The boundary atoms are fixed [35]. Atomic motion occurs in the thermostat and Newtonian atoms, and these two parts obey the classical second Newton’s law, which are integrated using the Velocity–Verlet algorithm [36] with a time step of 0.01 ps. NVE ensemble is used in the simulation, and the thermostat atoms are set to ensure reasonable outward heat conduction using the direct velocity scaling method. Initially, velocities of atoms, except fixed boundary ones, are set randomly under an equivalent temperature *T* = 300 K. The simulation process is divided into three phases: relaxation, loading and sliding. Before loading, the system is relaxed to reach its minimum energy configuration. Then, the tip is pressed to the substrate to apply the load, and the tip slides along *x* direction after loading.

An embedded-atom method (EAM) potential is employed to describe interactions between all the atoms, and the total energy, *U*_tot,_ is given by:(1)$$U_{\mathrm{tot}}=\sum_{i}U_{i}$$
where the energy *U_i_* of an atom *i* is given by:(2)$$U_{i}=F_{i}\left(\sum_{j\neq i}\rho_{i}(r_{ij})\right)+\frac{1}{2}\sum_{j\neq i}\phi_{ij}(r_{ij})$$
where *F_i_* is the embedding energy of atom *i*, *ρ_i_* is the electron density at site *i* induced by all other atoms in the model, and *ϕ_ij_* is a pair potential interaction between atoms *i* and *j*.

The force of atom *i* in *x*, *y* and *z* directions can be obtained as,
(3)$$\left\{ \begin{array}{c} F_{x}=-\frac{\partial U_{i}}{\partial x}=-\left(\frac{\partial F_{i}}{\partial x}+\frac{1}{2}\frac{\partial \phi_{ij}}{\partial x}\right)\\ F_{y}=-\frac{\partial U_{i}}{\partial y}=-\left(\frac{\partial F_{i}}{\partial y}+\frac{1}{2}\frac{\partial \phi_{ij}}{\partial y}\right)\\ F_{z}=-\frac{\partial U_{i}}{\partial z}=-\left(\frac{\partial F_{i}}{\partial z}+\frac{1}{2}\frac{\partial \phi_{ij}}{\partial z}\right) \end{array}\right.$$

Given the initial coordinates, velocities, and accelerations of all the atoms at time *t*, these parameters can be calculated by Velocity–Verlet algorithm [36] at time *t* + Δ*t*,
(4)$$\left\{ \begin{array}{l} r(t+\Delta t)=r(t)+v(t)\Delta t+\frac{1}{2}a(t)\Delta t^{2}\\ v(t+\Delta t)=v(t)+a(t)\Delta t\\ a(t+\Delta t)=-\frac{1}{m}\nabla U(r(t+\Delta t)) \end{array}\right.$$
where Δ*t* is time step, **r** is coordinate vector, **v** is velocity vector, **a** is acceleration vector, and *m* is mass of an atom.

## 3. Results and Discussion

### 3.1. The Influence of Textured Surface on Sliding Contacts

The textured surfaces influence the contact areas, and further influence the friction behaviors. In this work, the friction properties of different textured surfaces are studied, including different depths (*d* = 0.3615, 0.723, 1.0845 nm), different widths (*w* = 0.3615, 0.723, 1.0845 nm), different orientations (*θ* = 0°, 30°, 45°, 60°, 90°), and different shapes (#-shape, rectangular, v-shape). Before sliding, the tip is pressed into substrate for 0.3615 nm. The sliding velocity is 5 m/s and sliding distance is 15 nm.

#### 3.1.1. Effects of Texture Depth

A smooth surface and three textured surfaces with different depths are shown in Figure 2. The texture width is *w* = 0.723 nm, and the spacing *g* = 1.0845 nm.

Figure 3 shows the average friction forces of the smooth surface and textured surfaces. Compared with the smooth surface, textured surfaces can reduce average friction forces effectively, and the average friction forces decrease with an increase in texture depth, which is consistent with the conclusions of Refs. [10,11,19,22].

For a friction force, there are adhesion and ploughing components [37]. The textured surface can reduce the adhesion and ploughing components, which contributes to improving friction behaviors. On the one hand, due to the spacings between the textures, the textured surface can reduce the contact area and the interfacial shear strength, which reduces the adhesion component. On the other hand, the ploughing component is influenced by the total volume of ploughing and the force necessary to displace unit volume [38]. With the increase in texture depth, more abrasive particles are trapped by the spacing, and then less atoms accumulate in front of the tip in the sliding process, which reduces the ploughing component. As a result, the two components of the friction force are reduced by the textured surface, so the average friction forces of the textured surfaces are lower than those in the smooth surface.

Actually, the depth of texture is not the greater the better, and it should be equal to the normal load. It aims to trap the abrasive particles, to reduce the amount of atoms stacked in front of the tip and reduce the ploughing component. If the depth of the texture is too high, the carrying capacity will be decreased, which may affect the other properties of the material.

#### 3.1.2. Effects of Texture Width

At the atomic scale, adhesion effects play important roles in the friction force, and the adhesion force depends on the contact area. Texture width affects the contact area between the tip and substrate, and further influences friction behaviors. Three textured surfaces with different widths are shown in Figure 4. The widths are *w* = 0.3615 nm, 0.723 nm and 1.0845 nm, and the corresponding texture densities are 20%, 40% and 60%, respectively. For the three textures, *w* + *g* = 1.8075 nm is a constant, and the texture depth is *d* = 0.723 nm.

Figure 5 shows the average friction forces of textured surfaces with different widths. With an increase in width, the average friction forces become higher, and the textured surface with the width *w* = 0.3615 nm has the lowest friction force in this work. In [19], the textured surface with smaller spacing has higher friction forces, similar to the bigger texture width, and both the smaller spacing and bigger texture width induce a higher texture density. The increase in texture width leads to an increase in contact area, which increases the adhesion component. Besides, the sliding process causes the fracture of the bonds, the transfer of abrasive particles and the generation of dislocations, and all these cause atom accumulation. Taking the case of *w* = 0.723 nm as an example, the atoms’ distribution during the sliding contact are shown in Figure 6. At the initial stage, there are some atoms filled into the spacings of the textures due to the scratch of the tip, and a few atoms are piled up in front of the tip, as shown in Figure 6a. As the sliding occurs, many atoms accumulate in front of the tip at the intermediate stage in Figure 6b. In the final stage, more atoms are piled up, and no textures are left in Figure 6c, as the spacings are filled by the accumulated atoms. As texture width increases, more abrasive particles are piled up in front of the tip, as shown in Figure 7, which induces a higher scratching depth and increases the ploughing component. The corresponding adhesion component and ploughing component are presented in Figure 8. The two components are increased with the increase in texture width. Compared with the smaller texture width, the ploughing component plays a more important role with a larger texture width. As a result, the increase of both adhesion and ploughing components results in a higher average friction force for a larger width.

#### 3.1.3. Effects of Texture Orientation

The texture orientation *θ* is defined as the angle between the longitudinal direction of texture and *y* axis, as shown in Figure 9. The tip slides along *x* direction. In order to study the effects of texture orientation on friction behaviors, five different orientations (*θ* = 0°, 30°, 45°, 60° and 90°) are used. For all the textures, the width is *w* = 0.723 nm, the spacing is *g* = 1.0845 nm, and the depth is *d* = 0.723 nm.

Figure 10 shows the average friction forces of textured surfaces with different orientations. For the orientations of 0° and 90°, the average friction forces are greatly higher than the other cases. The case of 0° orientation presents the highest average friction force, and the average friction force of 45° orientation shows the minimum value.

In order to explain this phenomenon, we employ the Frenkel–Kontorova–Tomlinson (FKT) model to analyze the role of structural commensurability in friction behaviors. As shown in Figure 11, when two layers are commensurate, atoms of the top layer are spatially matched with the atoms of the bottom layer. The contribution of each atomic pair to the total force is maximized, thus leading to a high friction force. On the other hand, when the two layers are incommensurate, the friction force of an atomic pair may be cancelled out among the other pairs, thus leading to a low friction force [39]. Some earlier studies [40,41,42,43] have explained the strong orientation dependence of the friction, and low friction or even superlubricity can be obtained by changing the rotation angle between the tip and substrate. The most convincing demonstration to date came from the measurements by Dienwiebel et al. [41], in which a small graphite flake with different orientations was moved on a graphite surface using an AFM. The results showed that at two orientations (0° and 60°) corresponding to the friction peaks, the flake and substrate lattices were perfectly aligned, which revealed a dependence of friction on the orientation.

Single crystal copper is a face-centered cubic lattice, so the distribution density, arrangement and interatomic binding force of the single crystal copper are different for different crystal directions. The lattice orientation of the tip is the same as those of 0° textures. For the orientation of 0° and 90°, the contact atoms between the tip and substrate show commensurability and the average friction forces are higher than the other cases. For orientations of 30°, 45° and 60°, the lattice of the tip mismatches with the lattice of the substrate, which can prevent the simultaneous stick-slip motion of all atoms in the contact, and the friction forces show low values.

From the view of the adhesion component and ploughing component, the contact area and the scratching depth are changed with the variation of the texture orientation. For the case of 0° orientation, the tip slides perpendicular to the texture, and the discontinuous shorter contacts induce a smaller contact area at the initial stage. Due to the scratching of the tip, the atoms are rearranged and accumulated, as shown in Figure 12, which leads to a larger contact area and a higher scratching depth, so the corresponding adhesion component and ploughing component are enhanced greatly. When the texture orientation is 90°, the tip slides parallel to the texture, and the continuous adhesion in the contact junction induces a high adhesion component and a high average friction force. In Figure 12, the atom accumulation of the case of 45° texture orientation indicates that the scratching depth is lower, and the contact area is smaller, which produces a lower ploughing component and a lower adhesion component, so the average friction force is the lowest.

From the thermodynamic theory in [44], the friction force was proportional to the mean width of the real contacts in *y* direction. With an increase in the orientation angle, the mean width of the real contacts will decrease, if the geometry of this system remains the same, and the friction force should decrease gradually. Actually, the atoms of the substrate are moved and rearranged all the time during the sliding process, and the difference between the mean widths of the real contacts for the different orientations becomes negligible. From the adhesion and ploughing components in Figure 13, when the orientation is *θ* = 0°, the proportion of the ploughing component is greater than the cases of *θ* = 45° and *θ* = 90°. Besides, the two components of *θ* = 0° are higher than the other two cases. For the cases of *θ* = 45° and *θ* = 90°, the adhesion component plays the main role in friction force, which shows in the influence of the texture orientation.

#### 3.1.4. Effects of Texture Shape

To investigate the effects of texture shape, three textured surfaces with different texture shapes are modeled, including #-shape, rectangular and v-shape, as shown in Figure 14. For the #-shape texture, the width of the horizontal or the longitudinal texture is *w* = 0.3615 nm, and the spacing is a 1.446 nm × 1.446 nm rectangular dimple. For the rectangular and v-shape textures, the width is *w* = 0.723 nm, and the spacing is *g* = 1.0845 nm. The depth is *d* = 0.723 nm, and the density is 40%.

Figure 15 shows the average friction forces of textured surfaces with different texture shapes. Compared with the effects of the depth, width and orientation, the effects of shape on friction behaviors are slight. The #-shape textured surface gets higher shear strength than the other two cases due to its combination of longitudinal and horizontal structure, and its average friction force is the highest. The #-shape textured surface here is similar to the rectangular dimpled textured surface in Ref. [16], and the rectangular dimpled textured surface showed a higher average friction force than the rectangular groove textured surface. For the rectangular and v-shape textured surfaces, as the sliding of the tip, the atoms are filled into the spacing and accumulate in front of the tip. The similar surface structure and the same orientation means that there is little difference between the contact areas and scratching depths, which leads to little difference between the average friction forces.

### 3.2. The Influence of Subsurface Defect on Sliding Contact

In an ideal lattice, atoms are located strictly regularly and periodically. In an actual lattice, due to the formation conditions of crystals, the thermal movement of atoms, or other reasons, there are some defects in crystals, and the symmetry of the lattice is not perfect, which changes the mechanical properties of the material. Besides, the defects can be produced during the sliding process, and produce a noticeable contribution to the friction [27].

A nanoscale sliding contact model with subsurface defects is shown in Figure 16. Three nanoscale ball defects are modeled, and the depth of the defect is represented by the distance *h* between the center of ball defect and surface. Five different locations are used to study the effects of defect position on friction behaviors, and the depths are *h* = 0.90375 nm, 3.615 nm, 7.23 nm, 9.94125 nm, and 12.6525 nm. To accommodate the defects, the dimensions of the substrate are 21.69 nm × 5.784 nm × 14.46 nm in *x*, *y* and *z* directions, respectively. The distance between adjacent balls in x direction is 5.4225 nm, and the radius of the ball defect is *r* = 0.723 nm. In *y* direction, the ball defects are located in the center of the model.

The average friction forces for different depths of ball defects are shown in Figure 17. For cases of perfect substrate, or where the defects are closer to the surface of the substrate, the average friction forces are increased with the increase in the depth of the defects. In this work, when *h* = 3.615 nm, the average friction force is the highest. When the depth is greater than 3.615 nm, the average friction forces decrease with the increase in depth, and the average friction forces gradually approach that of the perfect substrate, which means that the effects of subsurface defects on friction become slight when the depth reaches a certain value.

Due to the existence of the ball defects, the surrounding atoms reach a new equilibrium state to produce the lattice distortion, which changes the material properties. In addition, compared with the perfect substrate, the defects make the sliding process unstable, and atoms near the defects of the substrate collapse during the sliding process. For the substrate contains defects, the common neighbor analysis shows that the dislocations are more than those of perfect substrate, as shown in Figure 18. With the increase in dislocation density, the shear strength of material increases, which increases the dissipation energy required to break the bond between atoms during the sliding process, and the friction forces are increased when *h* ≤ 3.615 nm.

In macro-contact mechanics, the stress distribution of two contact bodies showed that the maximum stress occurred at the subsurface. There was a similar phenomenon in nanoscale contacts, and the position of maximum stress showed a certain distance from the surface [45]. This distance depends on the radius of the contact body, the applied load and the material properties. If the defects are located near the position of the maximum stress, or they coincide with each other, the influence of the defects on the sliding process will be greater. On the other hand, if the defects are far away from the position of the maximum stress, the effects will be slight, and this could be the reason that the average friction forces become lower with the increase in depth of the defects, when *h*>3.615 nm.

## 4. Conclusions

Using an EAM potential, MD simulations have been carried out to study the effects of FCC copper textured surfaces with different depths, widths, orientations and shapes on friction behaviors during nanoscale sliding contacts. The influence of subsurface defects on the substrate is also discussed. Under the conditions and material used in this paper, some conclusions are drawn, as follows:(1)Compared with a smooth surface, textured surfaces can reduce friction forces effectively. In addition, the textured surfaces with a higher depth and smaller width lead to lower average friction forces.(2)For the texture surfaces with the orientations of 0° and 90°, the average friction forces are higher than those with orientations of 30°, 45° and 60°, and the textured surface with 45° orientation produces the lowest average friction force.(3)The effects of texture shape on friction behaviors are slight compared with the effects of depth, width and orientation, and the v-shape shows a lower average friction force.(4)The nanoscale ball defects in the substrate make the sliding process unstable. In this paper, when the depth of the ball defects is *h* = 3.615 nm, the influence of defects on the average friction force is the greatest. When the depth is greater than 3.615 nm, the average friction forces decrease with the increase in depth, and they tend to be close to that of a perfect substrate.

## Figures and Tables

**Figure 1 nanomaterials-09-01617-f001:**
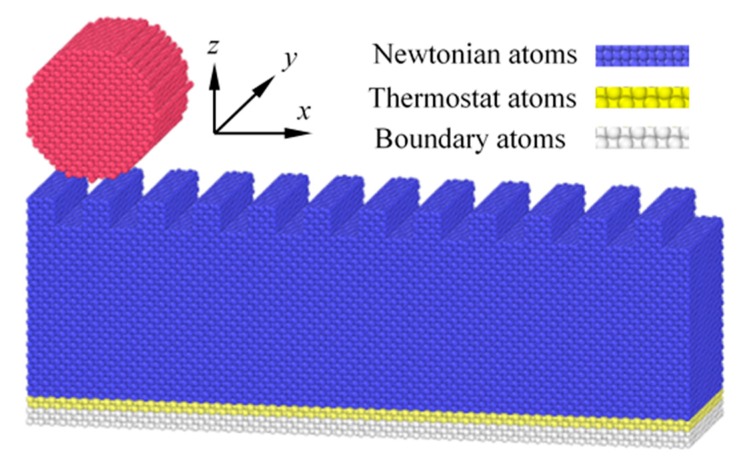
MD model of the sliding contact.

**Figure 2 nanomaterials-09-01617-f002:**
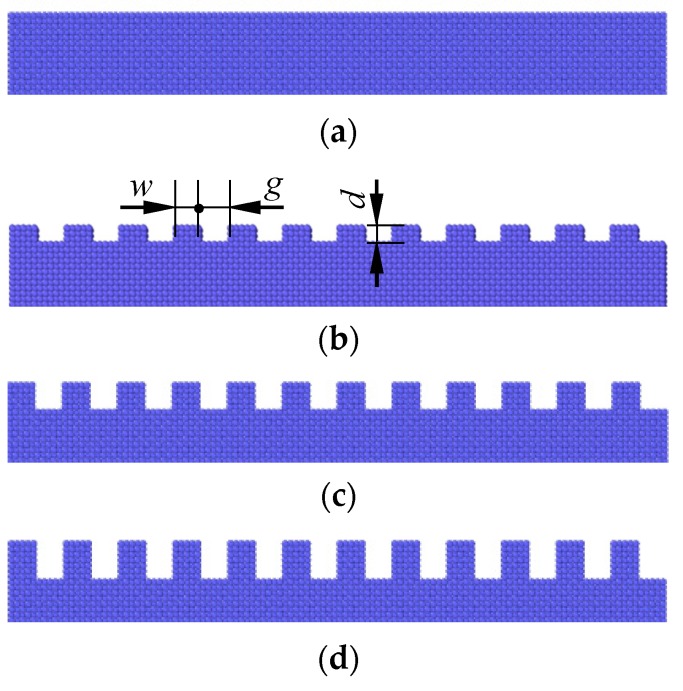
Textured surfaces with different depths. (**a**) Smooth surface; (**b**) *d* = 0.3615 nm; (**c**) *d* = 0.723 nm; (**d**) *d* = 1.0845 nm.

**Figure 3 nanomaterials-09-01617-f003:**
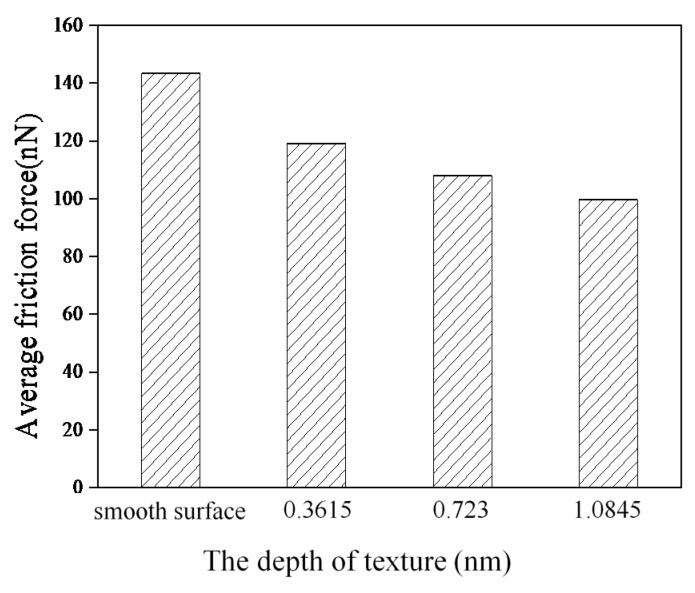
Average friction forces of the smooth surface and textured surfaces with different depths.

**Figure 4 nanomaterials-09-01617-f004:**
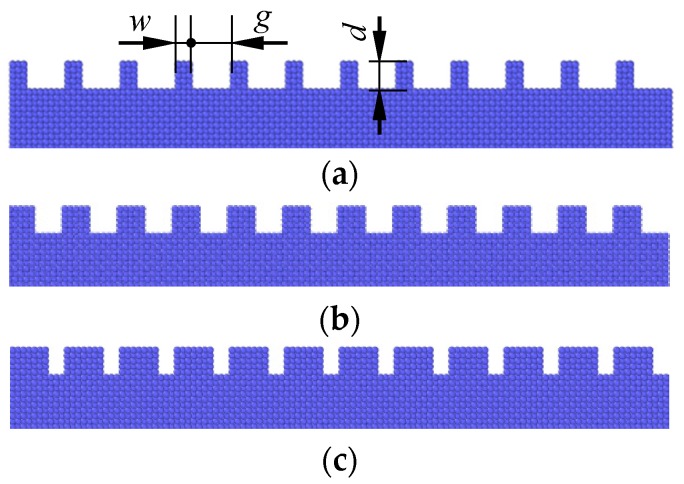
Textured surfaces with different widths. (**a**) *w* = 0.3615 nm; (**b**) *w* = 0.723 nm; (**c**) *w* = 1.0845 nm.

**Figure 5 nanomaterials-09-01617-f005:**
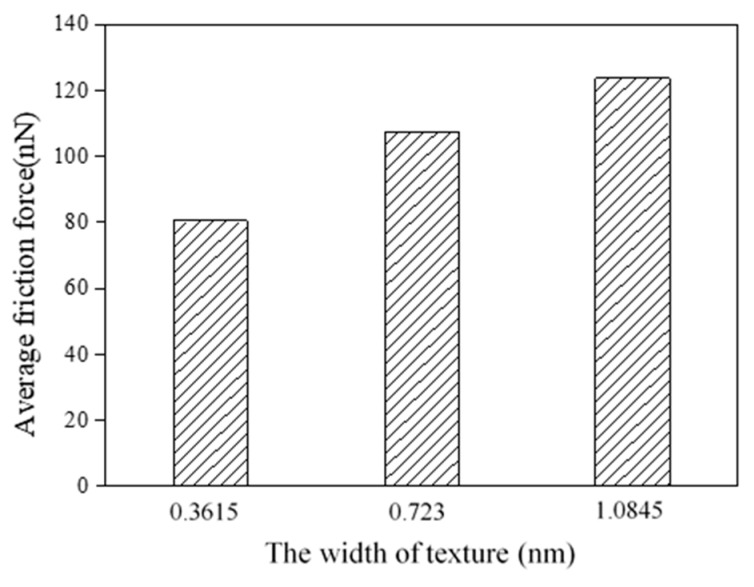
Average friction forces of textured surfaces with different widths.

**Figure 6 nanomaterials-09-01617-f006:**
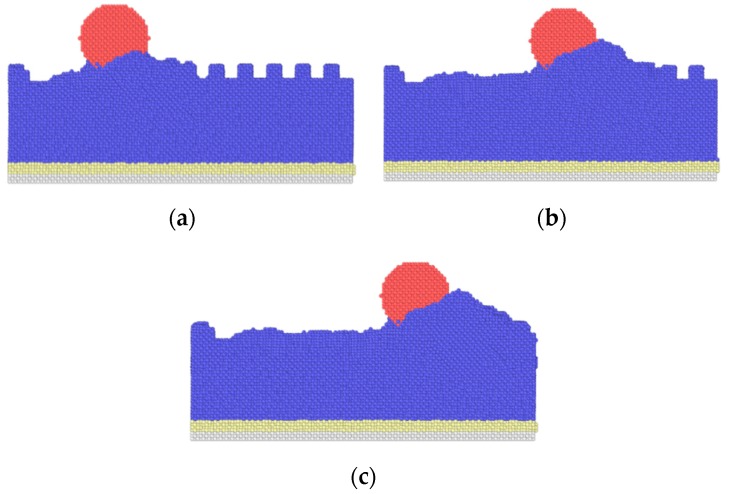
Atoms distributions during sliding process (*w* = 0.723 nm). (**a**) initial stage; (**b**) intermediate stage; (**c**) final stage.

**Figure 7 nanomaterials-09-01617-f007:**
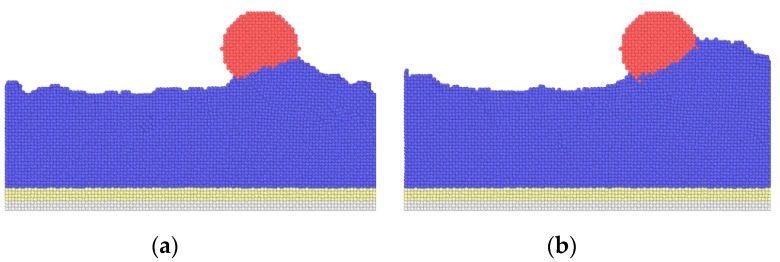
Atom accumulation in sliding process. (**a**) *w* = 0.3615 nm; (**b**) *w* = 1.0845 nm.

**Figure 8 nanomaterials-09-01617-f008:**
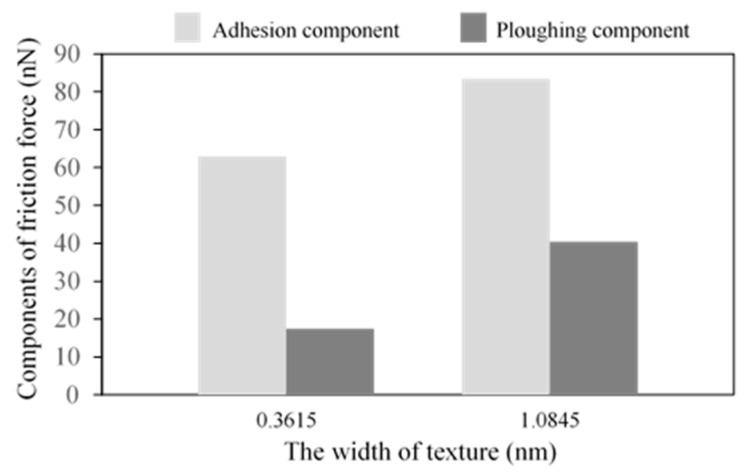
Components of friction force.

**Figure 9 nanomaterials-09-01617-f009:**
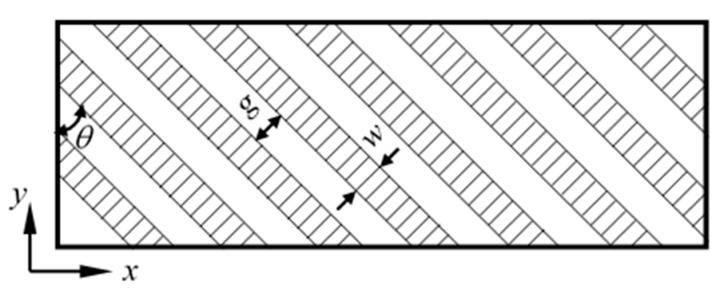
The definition of the texture orientation.

**Figure 10 nanomaterials-09-01617-f010:**
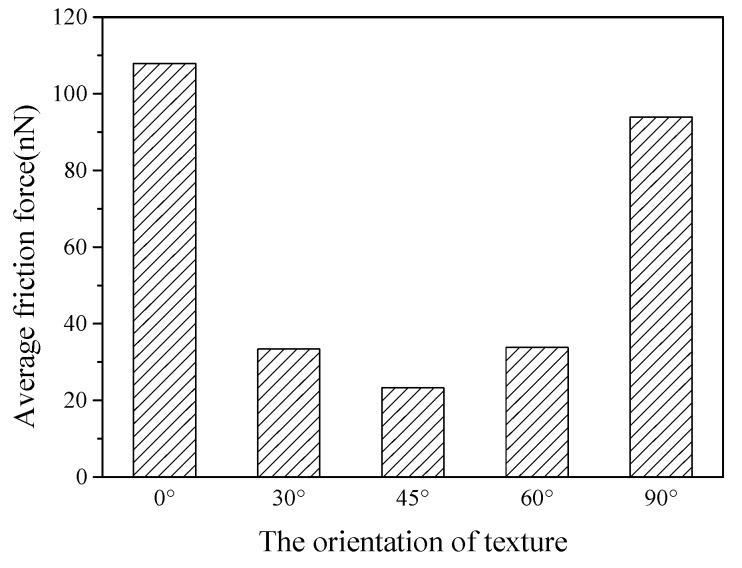
Average friction forces of textured surfaces with different orientations.

**Figure 11 nanomaterials-09-01617-f011:**
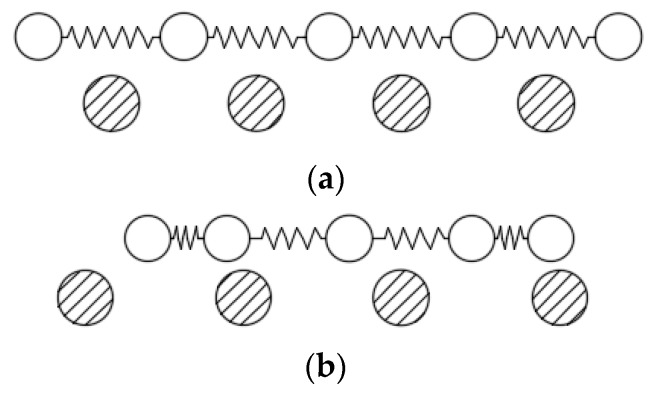
Schematic of the Frenkel–Kontorova–Tomlinson (FKT) model [39]. (**a**) Commensurate; (**b**) incommensurate.

**Figure 12 nanomaterials-09-01617-f012:**
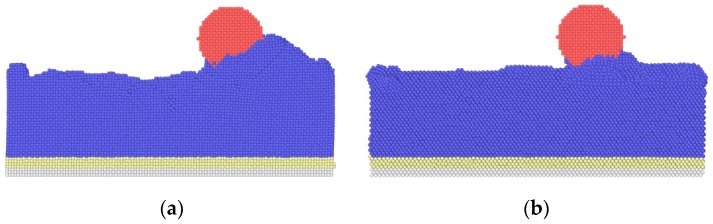
Atom accumulation in sliding process. (**a**) *θ* = 0°; (**b**) *θ* = 45°.

**Figure 13 nanomaterials-09-01617-f013:**
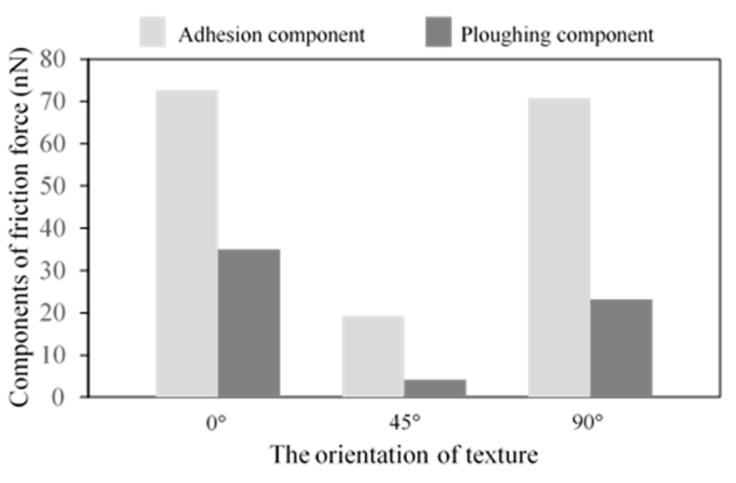
Components of friction force.

**Figure 14 nanomaterials-09-01617-f014:**
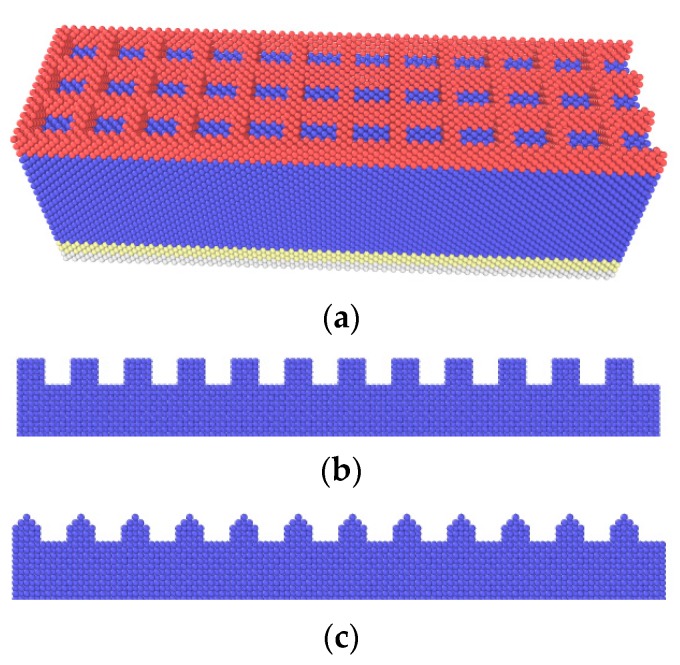
Textured surfaces with different shapes. (**a**) #-shape; (**b**) rectangular; (**c**) v-shape.

**Figure 15 nanomaterials-09-01617-f015:**
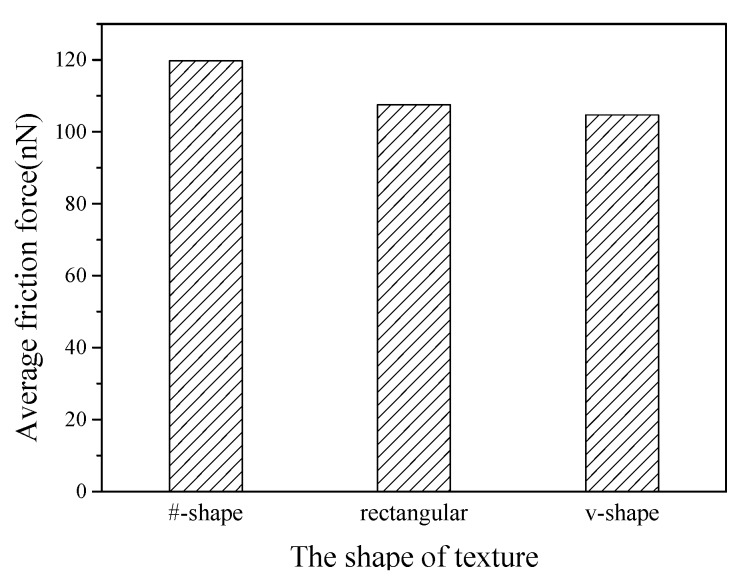
Average friction forces of textured surfaces with different shapes.

**Figure 16 nanomaterials-09-01617-f016:**
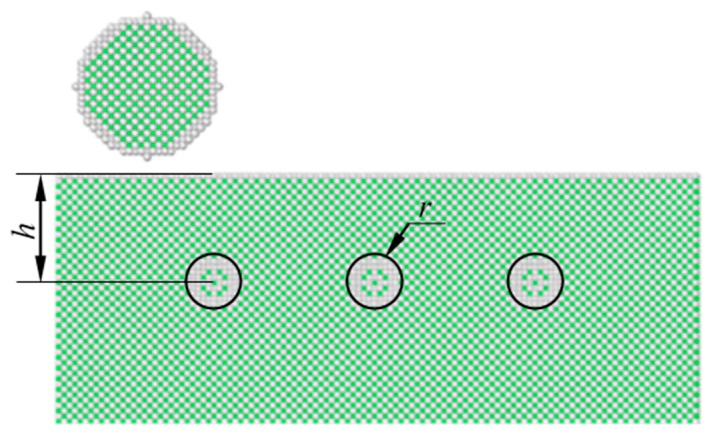
Sliding contact model with subsurface defects in substrate.

**Figure 17 nanomaterials-09-01617-f017:**
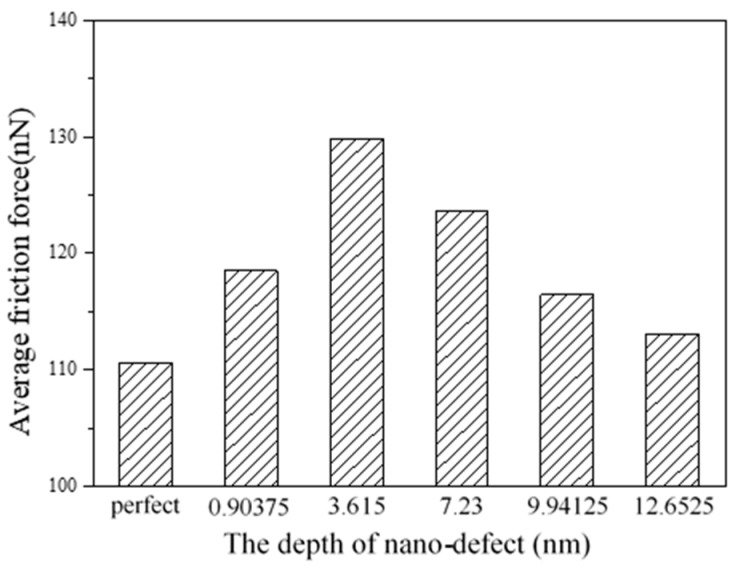
Average friction forces for different depths of defects.

**Figure 18 nanomaterials-09-01617-f018:**
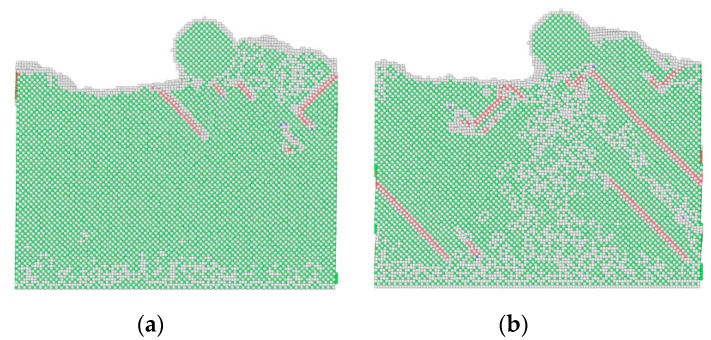
The dislocation of substrate during sliding process. (**a**) Perfect substrate; (**b**) substrate with defects (*h* = 3.615 nm).

**Table 1 nanomaterials-09-01617-t001:** Simulation parameters.

Parameter	Value
material	single crystal copper
dimension of substrate	21.69 nm × 5.784 nm × 7.23 nm
radius of tip	2.169 nm
potential	EAM
time step	0.01 ps
sliding velocity	5 m/s
sliding distance	15 nm
texture depth	0.3615 nm, 0.723 nm, 1.0845 nm
texture width	0.3615 nm, 0.723 nm, 1.0845 nm
texture orientation	0°, 30°, 45°, 60°, 90°
texture shape	#-shape, rectangular, v-shape
depth of subsurface defects	0.90375 nm, 3.615 nm, 7.23 nm, 9.94125 nm, 12.6525 nm

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
