# Peer review of "Influence of Nanoscale Textured Surfaces and Subsurface Defects on Friction Behaviors by Molecular Dynamics Simulation"

_nanomaterials, 2019, doi:10.3390/nano9111617_

Round 1

Reviewer 1 Report

The authors present classical molecular dynamics simulations to describe friction. The model consisting of several thousand atoms is interesting. The manuscript is well written, but a language check could improve it.

Some questions/comments:
- In the introduction the authors mention all kinds of materials. Then they do computations for a copper model only. In the conclusions this limitation is not mentioned. Please comment on the transferability of the model.
- The cylinder tip is assumed to be rigid. Does such a model still describe copper reliably? In future simulations one might put the temperature in the tip to zero instead
of completely fixing the atoms.
- What about oxidation and hydroxylation? For every metal, except maybe gold, the surface will be oxidized, which greatly influences the amount of adhesion.
- The next interesting step could be addition of water or a lubricant. I would assume that suitable force fields exist?

Reviewer 2 Report

This is a straightforward paper attempting to demonstrate texture morphology effects on friction in nanoscale by MD simulations. My main concerns are the following:

1) The connection of the references, given in the Introduction, to this study is somewhat unclear. Some of them are for lubricated cases and some of microscale. A better presentation of the results in the literature that can be used to verify these results is necessary. 

2) In Figs. 6 and 10 the scale appears to be about the same as in Fig. 1. Why then, the texture is not visible in those figures, although it should be almost of the same scale as the cylinder radius? This makes the results difficult to understand. Also, more efforts should be done to evaluate the role of the ploughing component in the simulated force.

3) The paper includes an attempt to explain the results by the FKT-model. That model assumes that the "adhesion friction" occurs at the true nanoscale interface. However, the success of the thermodynamic theory of friction in explaining data (AIP Advances 2, 012179) shows that the frictional force arises at the edges of the true contacts. Interestingly, the texture scale in this manuscript is very close to the true contact size assumed in the thermodynamic theory. This would allow interpretations of these results in view of the themodynamic theory. Since in that theory a longer true contact length reduces the frictional force, this may not be easy. Probably, one should conclude that the ploughing component is dominating here. A quantitative estimate of that is necessary.     

Reviewer 3 Report

The topic of this manuscript falls within the scope of Nanomaterials.

The paper is concerned with the influence of nanoscale textured surfaces and subsurface defects on friction behaviors. The authors use an approach based on molecular dynamics. The reviewer finds no fault whatsoever with the methods, numerical analysis, or conclusions. The work is fundamentally sound. The reviewer would recommend publishing it.

Author Response

Dear reviewer:

Thanks for your work and recommendations, and we list the responses as follows.The purple revisions are for the grammatical errors, typos, language usage, and some other revisions.

P1 line 16, “structure” is changed to “structures” P1 line 21, “and” is changed to “or” P1 line 38, “the” is changed to “a” and “micro/” is deleted P2 line 54, “texture” is changed to “textures” P2 line 70, “simulation” is changed to “simulations” P2 line 81, “simulation” is changed to “simulations” P2 line 84, “the” is changed to “a” P2 line 85, “decrease” is changed to “decreased” P2 line 87, “changes” is changed to “change” P3 line 98, “When defects accumulated” is changed to “When defects were accumulated” P3 line 103, “simulation” is changed to “simulations” P3 line 105, “occurred” is changed to “was presented” P3 line 107, “simulation” is changed to “simulations” P3 line 130, “divided to” is changed to “composed of” P3 line 132, “the” is changed to “these” P5 line 163, “The” is changed to “A” P6 line 194, “affects” is changed to “may affect” P6 line 199, “width is” is changed to “widths are” P7 line 218-219, “The larger the texture width is, the more abrasive particles piled up in front of the tip” is changed to “As the increase of the texture width, more abrasive particles are piled up in front of the tip” P7 line 223, we add “As a result,” P7 line 233-234, “The width of all the textures is w=0.723 nm, the spacing is g=1.0845 nm, and the depth is d=0.723 nm” is changed to “For all the textures, the width is w=0.723 nm, the spacing is g=1.0845 nm, and the depth is d=0.723 nm.” P9 line 299, we add “dimple” P10 line 329, “depth is” is changed to “depths are” P11 line 344, “comparing” is changed to “compared” P12 line 371, “Comparing” is changed to “Compared”

These revisions are highlighted in purple in the paper.

Round 2

Reviewer 2 Report

It is essential to visually understand what happens with the asperities during the sliding. Hence, it would help a reader if the figure [R1] that the authors present in their response document is included in this paper. Perhaps, it could replace some of the present figures.
